# CFD Simulation Analysis on Make-up Air Supply by Distance from Cookstove for Cooking-Generated Particle

**DOI:** 10.3390/ijerph17217799

**Published:** 2020-10-25

**Authors:** Hyungkeun Kim, Kyungmo Kang, Taeyeon Kim

**Affiliations:** 1Department of Architectural Engineering, Yonsei University, Seoul 03722, Korea; hang0621@hanmail.net; 2Department of Living and Built Environment Research, Korea Institute of Construction Technology, Goyang-Si 10233, Korea; kyungmokang@kict.re.kr

**Keywords:** indoor air quality, cooking-generated particle, ventilation, range hood, CFD simulation

## Abstract

Indoor cooking is the main cause of particulate matter (PM) within residential houses along with smoking. Even with the range hood turned on, cooking-generated PM can spread quickly into the living room due to the heat generated by the cookstove. In order to improve the PM spread prevention performance of the range hood, a supply of make-up air is needed. Generally, make-up air is supplied through a linear diffuser between the kitchen and living room. In such cases, it is necessary to determine the appropriate location of the supply diffuser. This study evaluates the spread of PM according to different locations of the supply diffuser, which feeds in make-up air. For this purpose, indoor airflow and PM spread were analyzed through CFD (Computational Fluid Dynamics) simulation analysis. By changing the location of the supply diffuser from the contaminant source, PM concentration was analyzed in the kitchen and living room of an apartment house in Korea. Based on the results, the optimal installation location was determined. In this study, 1.5 m from the source was the most effective location of make-up air supply to prevent the spread of cooking-generated particles.

## 1. Introduction

Residents spend 90% of their time indoors [1]. As a result, their health is greatly affected by indoor air quality [2]. When residents are exposed to poor indoor air quality, they can suffer from respiratory and skin diseases, among others. Among indoor air pollutants, particulate matter (PM) enters the human body through the respiratory tract and can cause various diseases [3]. Indoor PM is caused by the inflow of outside air through active and passive ventilation and by residents’ activities such as smoking, cooking and cleaning [4]. Smoking and cooking significantly contribute to increase indoor PM concentrations [5]. Cooking is the main source of PM in nonsmoking households [6,7]. In particular, high cooking temperatures can speed up the spread of PM [8]. Therefore, in addition to the kitchen, where cooking takes place, the spread of PM into adjacent spaces such as the living room may be a problem.

Operating the range hood is the most efficient method for removing indoor PM [9]. PM emissions from cooking can be rapidly removed by operating the range hood [10]. In particular, PM generated during cooking can be effectively removed by opening any windows while operating the range hood [11]. If make-up air is not supplied through natural or mechanical ventilation while operating the range hood, the exhaust efficiency may decrease [12]. A separate supply of make-up air is required to solve this issue. In order to provide make-up air to the range hood, it is effective to install the most effective auxiliary supply system that prevents the spread of pollutants [13]. The auxiliary supply system is usually installed in the vicinity of the range hood to prevent the spread of contaminants. The auxiliary supply system supplies outside air through a line diffuser to avoid the spread of contaminants along the ceiling [14]. The effect of preventing the spread of contaminants may vary depending on the location of the diffuser. Therefore, it is necessary to install the diffuser in an optimal location.

This study analyzes the effect of the location of make-up air installation on the prevention of the spread of PM generated in the kitchen. Under the condition that the range hood and line diffuser for supplying make-up air are operated simultaneously, the extent of PM spread was analyzed. A case study on typical flat-type apartments in Korea was carried out through CFD simulation and analysis. The optimal location for the auxiliary supply system is derived based on the results.

## 2. Building Description and Field Measurement

### 2.1. Building Description

CFD simulation analysis was conducted based on an experimental apartment of a company located in Pyeongtaek, Gyeonggi-do. This house has a typical Korean apartment floor plan. The floor area is 92.57 m^2^, which is considered a medium-sized apartment in Korea. The ceiling height is 2.3 m, which corresponds to that of a standard apartment. This study only analyzed the living room and kitchen. The area of the living room and kitchen is 47.01 m^2^ (Figure 1).

Although the experimental apartment was designed to perform ventilation through a Heat Recovery Ventilator (HRV), the ventilation was deactivated for the purpose of this analysis, which was to evaluate the performance of the auxiliary supply system. A range hood is installed in the kitchen and a supply diffuser is installed to support the range hood. The auxiliary supply system is located on the ceiling about 1.5 m from the range hood and the length of the linear diffuser is 0.9 m. In this study, to validate the CFD simulation model, the range hood and auxiliary supply system were operated with a wind volume of 150 CMH (m^3^/h), respectively. Measurements were taken at a height of 1.5 m each inside the kitchen living room and intermediate position (Figure 2). In this study, the simulation model was verified by comparing the measurement results. To perform the simulation validation, the temperature distribution by height was measured at points P1 and P2. In P1, P2 and P3, the concentration of carbon dioxide was measured by placing small tubes.

### 2.2. Field Measurement for Validation of CFD Model

Temperature and carbon dioxide concentration were measured to validate the CFD simulation model. Cooking was conducted to determine the amount of contaminants released and the PM 2.5 concentration was measured. In addition, K- and T-type thermocouples were installed to measure the temperature at various points and data were collected with a data logger (SATO SK-L200, Tokyo, Japan). Temperatures were measured at each wall, floor and ceiling of the experimental apartment and above the cooktop. In particular, the temperature by height was measured in the center of the living room and occupant position in the kitchen to compare the temperature distribution by height (0.1, 0.3, 0.5, 1.0, 1.5, 2.0 and 2.2 m). All measurement data were averaged.

As for the cookstove, an induction type was used because temperature control is convenient and the risk of ash generation due to combustion is removed. A round-shaped pot was placed on top of the cookstove. To measure the surface temperature of the pot during cooking, K-type thermocouples were attached to the center of the pot. Fine particles are difficult to release consistently until the indoor concentration reaches a steady state. Carbon dioxide is widely used as a tracer gas when evaluating range hood performance inside a kitchen. In this study, 10 L/min of carbon dioxide was released through a flow meter in the laboratory. Carbon dioxide was discharged vertically downwards onto the round-shaped pot (Figure 3). Carbon dioxide concentration was measured with an INOVA multipoint sampler. The concentration of carbon dioxide was measured by placing a small tube at a height of 1.5 m in the locations of Points 1, 2 and 3.

## 3. Description for CFD Simulation

Methods for evaluating indoor air quality include experimental and simulation analysis. Laboratory measurement is the most accurate method, but it is expensive to set up a laboratory and purchase test equipment. It is difficult to understand the spatial variations because only data from the measuring point can be obtained [15]. CFD is most often used for indoor air quality simulation analysis [16]. It enables analyzing various cases that are difficult to analyze in real situations [17,18]. Unlike laboratory measurements, CFD can provide physical data for the entire space [19]. This study analyzed the effect of the range hood and the provision of make-up air by using CFD. Concentration was analyzed by varying the distance between the diffuser of the auxiliary supply system and the cookstove from 1.0 to 2.5 m at 0.5 m intervals (Figure 4). The amount of PM emission during cooking was calculated by dividing the value measured at the exhaust outlet of the range hood by time.

### 3.1. Numerical Methods

In this study, indoor space was analyzed using Star-ccm+, a commercial CFD software developed by Siemens. Star-ccm+ is suitable for analyzing physical problems based on the finite volume method. Star-ccm+ has been verified for academic applicability because it has been used in numerous research projects regarding the thermal and air environment of indoor spaces [20,21]. Recently, it has also been widely used in ultrafine particle analysis [22,23]. To discretize the Navier–Stokes equation, the SIMPLE algorithm was used. Autodesk’s AutoCAD was used for modeling the indoor space, which was divided using polyhedral mesh for flow analysis.

Most PM generated during cooking has a particle size of 1 μm or less [24]. The dynamic behavior of PM is strongly related to particle size [25]. At a smaller PM size, its behavior is more affected by the Brownian diffusion than by the deposition [26]. When the PM size is larger, deposition occurs due to gravitational settling [7]. In particular, gravitational settling effects must be considered when analyzing PM movements with diameters larger than 2.5 μm [27]. Unlike other gaseous contaminants, PM shows different behaviors depending on the particle size due to the effect of gravity. Therefore, in order to analyze the behavior of PM, a multiphase model should be applied for calculations. To analyze the spread of indoor PM, the Eulerian multiphase model was applied.

In this study, the Reynolds-averaged Navier–Stokes (RANS) Turbulence model was employed. To obtain the RANS equations, each solution variable ϕ (velocity components, pressure, energy, species concentration, etc.) in the instantaneous Navier–Stokes equations is decomposed into its averaged value $\bar{\phi }$ and its fluctuating component $\phi^{\prime }$ Equation (1).
(1)$$\phi =\bar{\phi }+\phi^{\prime }$$

The averaging process may be regarded as time-averaging for steady-state situations and ensemble-averaging for repeatable transient situations. Inserting the decomposed solution variables into the Navier–Stokes equations results in equations for the mean quantities.
(2)$$\frac{\partial \rho }{\partial t}+\nabla \cdot \left(\rho \bar{v}\right)=O$$
(3)$$\frac{\partial }{\partial t}\left(\rho \bar{v}\right)+\nabla \cdot \left(\rho \bar{v}⊗\bar{v}\right)=-\nabla \cdot \bar{p}Ι+\nabla \cdot \left(T+{Τ}_{t}\right)+f_{b}$$
where ρ  is the density, $\bar{\nu }$ and $\bar{p}\;$ are the mean velocity and pressure, respectively, Ι is the identity tensor, Τ is the viscous stress tensor and $f_{b}$ is the resultant of the body forces (such as gravity and centrifugal forces).
(4)$$\left(T_{t}\right)=-\rho \left(\begin{array}{ccc} \bar{u^{\prime }u^{\prime }} & \bar{u^{\prime }v^{\prime }} & \bar{u^{\prime }w^{\prime }}\\ \bar{u^{\prime }v^{\prime }} & \bar{v^{\prime }v^{\prime }} & \bar{v^{\prime }w^{\prime }}\\ \bar{u^{\prime }w^{\prime }} & \bar{v^{\prime }w^{\prime }} & \bar{w^{\prime }w^{\prime }} \end{array}\right)$$

The standard k-ε low Reynolds number model is the low Reynolds number approach merged into the standard k-ε model. In this model, coefficients that are distinct from the standard k-ε model are applied. This model provides a more precise analysis of the viscous-affected regions at the near walls [28].

The turbulent eddy viscosity $\mu_{t}$ is calculated as Equation (5).
(5)$$\mu_{t}=\rho C_{\mu }f_{\mu }kT$$
where ρ  is the density, $C_{\mu }\;$ is model coefficient, $f_{\mu }\;$ is a damping function and T  is the turbulent time scale.

The transport equations for the kinetic energy k and the turbulent dissipation rate are as follows:(6)$$\frac{\partial }{\partial t}\left(\rho k\right)+\nabla \cdot \left(\rho k\bar{v}\right)=\nabla \cdot \left[\left(\mu +\frac{\mu_{t}}{\sigma_{k}}\right)\nabla k\right]+P_{k}-\rho \left(\varepsilon -\varepsilon_{0}\right)+S_{k}$$
(7)$$\frac{\partial }{\partial t}\left(\rho \varepsilon \right)+\nabla \cdot \left(\rho \varepsilon \bar{v}\right)=\;\nabla \cdot \left[\left(\mu +\frac{\mu_{t}}{\sigma_{\varepsilon }}\right)\nabla \varepsilon \right]+\frac{1}{T_{e}}C_{\varepsilon 1}P_{\varepsilon }-C_{\varepsilon 2}f_{2}\rho \left(\frac{\varepsilon }{T_{e}}-\frac{\varepsilon_{0}}{T_{0}}\right)+S_{\varepsilon }$$
where $\bar{v}\;$ is the mean velocity, μ  is the dynamic viscosity, $\sigma_{k},\;\sigma_{\varepsilon },C_{\varepsilon 1}$ and $C_{\varepsilon 2}\;$ are model coefficients, $P_{k}\;$ and $\;P_{\varepsilon }\;$ are production terms, $f_{2}\;$ is a damping function and $S_{k}\;$ and $S_{\varepsilon }\;$ are the user-specified source terms.

### 3.2. Boundary Conditions

For a reliable CFD simulation analysis, appropriate boundary conditions should be entered. Input data was measured in order to simulate the actual situation of the experiment. The input data include surface temperature, air temperature (supply air), velocity and particle emission rate.

In order to analyze the indoor PM, it is necessary to know the concentration of the PM that is emitted during cooking. TSI-8532 was installed in the duct outlet of the range hood to analyze the PM emission during cooking. The PM concentration generated during cooking was measured and accumulated, divided by cooking time and converted into the PM emission rate. In addition, the temperature of the frying pan was measured using a K-type thermocouple attached at the top of the cookstove.

The measured emission rate of the PM and the surface temperature of the frying pan are as follows. The surface temperature of the frying pan was an average of 163 °C (range: 79.5–199.7 °C), and the temperature of the supply air was an average of 27.5 °C. The emission rate of PM was about 3.07 mg/s during cooking. The occupant heat source was 76 W referring to the ASHRAE (a standing person) [29]. All boundary conditions are described in Table 1.

## 4. Measurement and Validation of CFD Simulation

Accurate and detailed modeling is required for reliable CFD simulation analysis. In this study, the geometries of the kitchen and the living room were modeled in detail to simulate the same situations as the experimental conditions. In addition, a human body model was created to apply the influence of an occupant during cooking. Prior to the simulation analysis, grid sensitivity analysis and validation were carried out via the tracer gas method (carbon dioxide). In the measurement process, the range hood and auxiliary supply system were operated simultaneously (flow rate: 150 CMH).

### 4.1. Grid Sensitivity Analysis

Grid sensitivity analysis was performed before the simulation analysis. CFD simulation depends heavily on the size or shape of the grid. Therefore, a mesh size that is not affected by the shape or size of the grid needs to be selected. In this study, the grid sensitivity analysis was performed by applying three grid sizes (Table 2). The analysis was compared with the normalized value. Data comparisons were made using data from P1 that were located near the cookstove and could be directly influenced by strong buoyancy. The temperature and carbon dioxide concentration of the P1 data were used for comparison. All data were normalized by the data in the fine grid model. According to the grid sensitivity analysis, the medium-sized grid model was within a 5% margin of error compared to the fine grid model (Figure 5). Therefore, in this study, simulation analysis was performed using the medium-sized grid.

### 4.2. Measurement Result and Validation of CFD Model

Prior to the simulation analysis, the simulation was validated by comparing room temperature data with the carbon dioxide concentration data. Measurements were taken every 10 s for more than an hour, and to simulate the steady state, data obtained when the temperature and concentration distribution reached a constant state were used. Carbon dioxide concentration was measured in the living room, kitchen and the area in between.

Initially, the temperature above the cookstove was kept constant at 163 °C. Comparing the measured data with the CFD simulation of the temperature and humidity inside the kitchen and living room, it appears that the temperature shows a very similar characteristic with the actual data. (Figure 6). In the kitchen, strong buoyancy is caused by heat from the cookstove. Heat from the cookstove spreads to the living room along the ceiling. As a result, the higher the height in the kitchen, the higher the temperature. Because of the spread of heat, the higher the height in the living room, the higher the temperature. This trend is shown in the CFD results as well as measurement data from the living room and kitchen. The analysis results of CFD tend to underestimate the thermal spread of the ceiling. Although absolute values of temperature between CFD and measured data differ by up to 1.7°, the temperature trend is similar by height. Therefore, this CFD model is considered suitable for assessing indoor environments.

Carbon dioxide also shows a distribution comparable to that of the actual measurement (Figure 7). The measured concentrations of P1, P2 and P3 are 582, 561 and 567 ppm, respectively. The CFD data are 576, 564 and 567 ppm, respectively. Although the concentration of carbon dioxide at each measuring point differs by up to 6 ppm, this is considered negligible for indoor air quality analysis. Therefore, it is determined that the simulation model developed in this study is suitable for conducting the case study.

## 5. Result and Discussion

### 5.1. Result of CFD Simulation

During the cooking process, strong buoyancy is generated by high-temperature heat from the cookstove [30]. Due to this buoyancy, the heat and contaminants that were not discharged through the range hood can spread into adjacent spaces along the ceiling. As the distance from the contaminant source increases, the speed of the spread into the living room decreases due to the friction on the ceiling. Supplying make-up air through a line diffuser has a similar effect as an air curtain, which can block the transfer of contaminants and heat into the living room.

Figure 8 and Figure 9 show the results of the CFD simulation analysis of PM 2.5. As the distance of the supply diffuser from the contaminant source increases, the PM concentration inside the kitchen shows a slight decrease. Looking at the distribution of the vertical section, as the distance of the supply diffuser decreases from the contaminant source, it can be seen that the direction of airflow from the ceiling to the living room becomes curved. As mentioned earlier, the closer the contaminant source, the faster the transmission speed; hence, it is inferred that the discharge of make-up air from the supply diffuser is affecting the direction of airflow. According to the analysis, the speed of the spread on the ceiling becomes relatively low at 0.3 to 0.4 m/s from a distance beyond 1.5 m. Therefore, the effect of the spread speed diminishes at a distance farther than 1.5 m. Due to this, the PM in the kitchen seems to be partially spreading to the living room. As the supply diffuser is farther away from the contaminant source, the airflow is smoothly supplied vertically from the ceiling, reducing the impact of PM spreading into the living room.

Concentration distribution in the living room shows that concentration is the lowest when the supply diffuser is located 1.5 m away. Space is formed between the kitchen and the supply diffuser where PM can remain. This space widens as the distance from the contaminant source increases. As this space expands, there is a possibility for the PM to spread into the living room. Therefore, it is necessary to adequately design the spread speed and expansion space of PM. For the indoor space used in this study, the spread of PM generated during cooking is minimized at a distance of 1.5 m.

### 5.2. Discussion

The purpose of this study was to propose an optimal method for preventing the spread of PM generated during cooking into the living room. To achieve this, the optimal location between the kitchen and the living room to install a linear-shaped supply diffuser was analyzed. The overall effect of preventing the spread of PM generated during cooking is high if make-up air is supplied while the range hood is operating. Cooking-generated PM mostly spreads along the ceiling; thus, the spreading pattern can vary greatly according to the distance of the diffuser.

It is effective to have the supply diffuser within a certain distance range away from the contaminant source. Additionally, it is difficult to block the spread if the distance from the contaminant source is too close or far. Figure 10 shows the average concentration in the kitchen and living room. When the distance of the supply diffuser is too close, both the living room and kitchen show the highest concentration. The PM concentration in the kitchen shows a slight decrease as its distance from the diffuser increases. However, it can be seen that the concentration in the living room decreases up to a certain distance but increases thereafter.

Even if the PM concentration in the kitchen is somewhat high, it can be rapidly discharged if the range hood is operated after cooking is finished [31]. However, the PM that has already spread into the living room cannot be removed easily even if the range hood is turned on after cooking. Therefore, when planning to supply make-up air with the range hood, it is necessary to have a plan of minimizing PM spread into the living room.

## 6. Conclusions

The cookstove inside the kitchen produces high-temperature heat and a large amount of PM while cooking. The generated heat and PM can spread to the living room along the ceiling. By operating the range hood and supplying make-up air simultaneously, such a spread into the living room can be reduced. The performance of spread prevention may vary depending on the location of the supply diffuser for make-up air. Therefore, in order to prevent PM from spreading into the living room from the kitchen, the supply diffuser needs to be installed in an appropriate location based on simulation analysis. Other factors such as the direction of air supply, length of air supply device and airflow volume can also affect the prevention of the spread of PM. We expect that a relative comparison of these factors can be conducted through various analyses as well.

## Figures and Tables

**Figure 1 ijerph-17-07799-f001:**
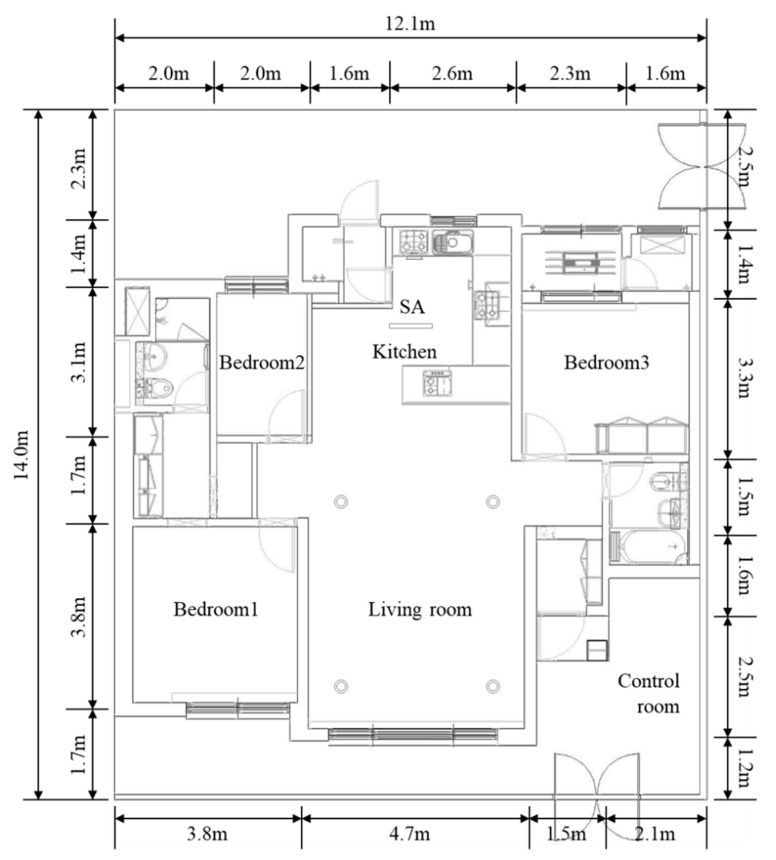
Floor plan of the target apartment.

**Figure 2 ijerph-17-07799-f002:**
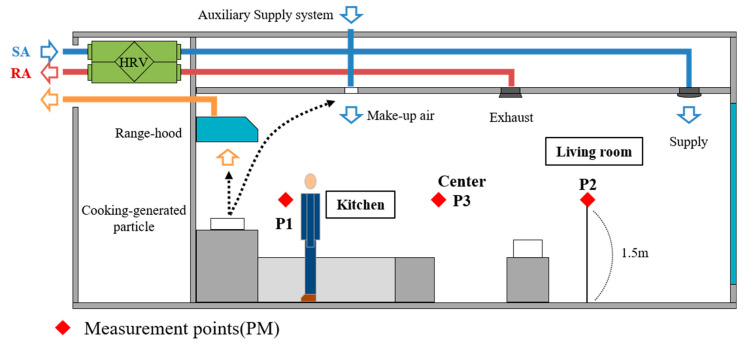
Section diagram of kitchen and living room.

**Figure 3 ijerph-17-07799-f003:**
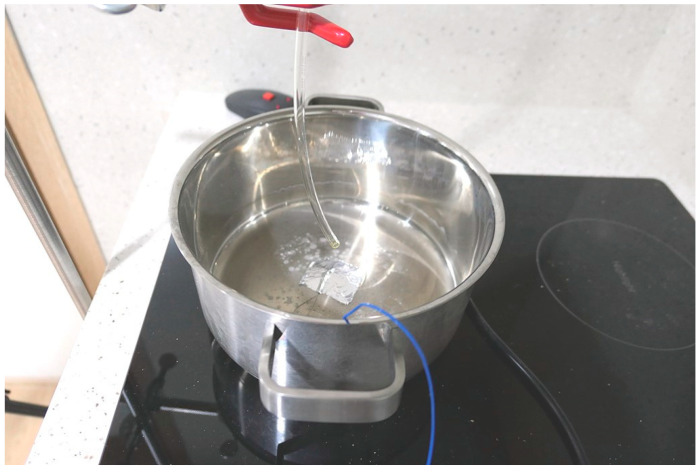
CO_2_ injection to the round-shaped pot.

**Figure 4 ijerph-17-07799-f004:**
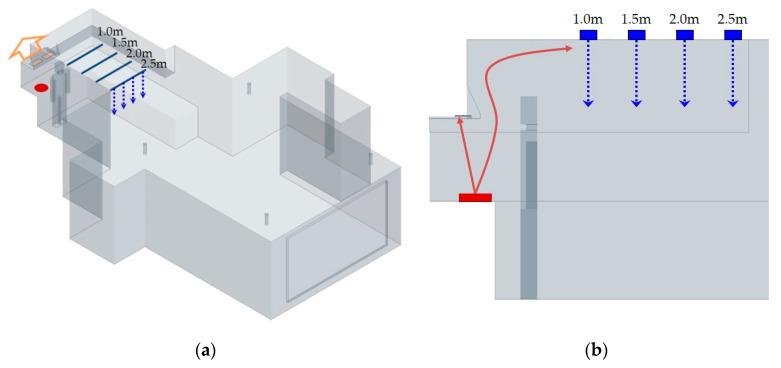
CFD simulation model and location of the supply air diffuser: (**a**) perspective diagram of kitchen hood and supply diffuser; (**b**) vertical section diagram of the location of the supply diffuser.

**Figure 5 ijerph-17-07799-f005:**
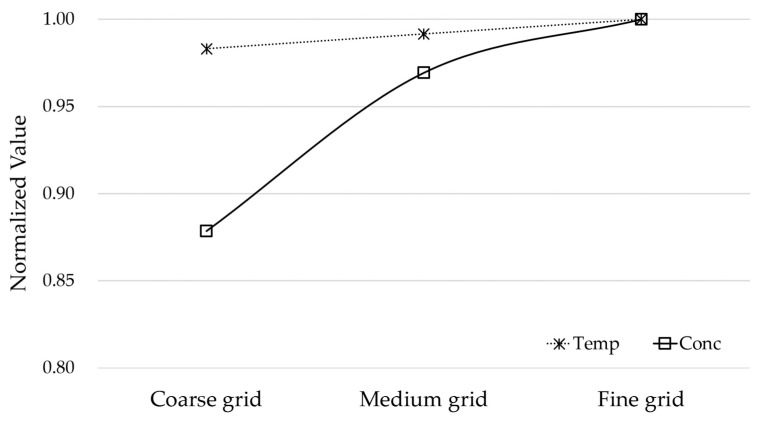
Result of the grid sensitivity analysis.

**Figure 6 ijerph-17-07799-f006:**
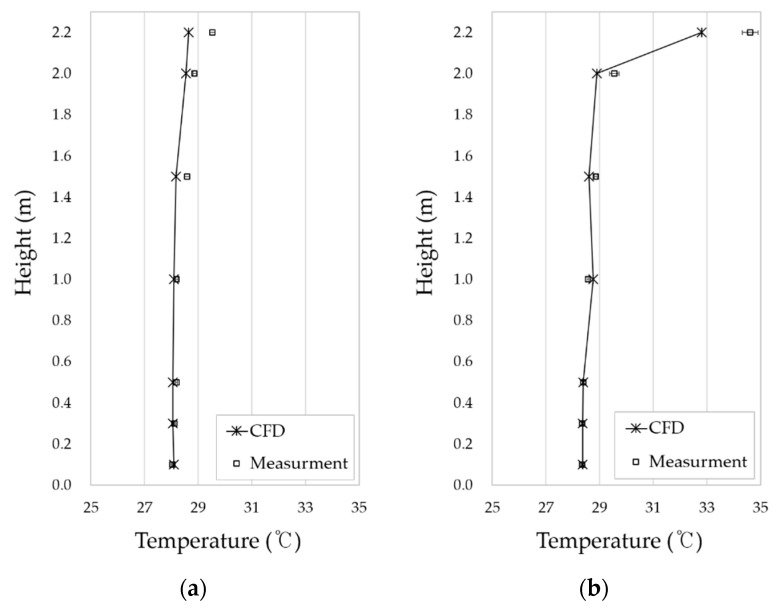
Temperature distribution of CFD and measuring data: (**a**) temperature distribution in the living room; (**b**) temperature distribution in the kitchen.

**Figure 7 ijerph-17-07799-f007:**
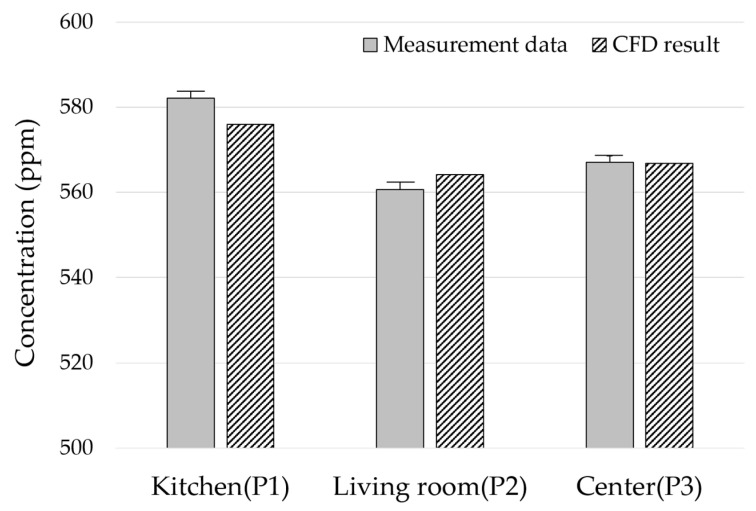
Comparison of CFD and measuring data (concentration of CO_2_).

**Figure 8 ijerph-17-07799-f008:**
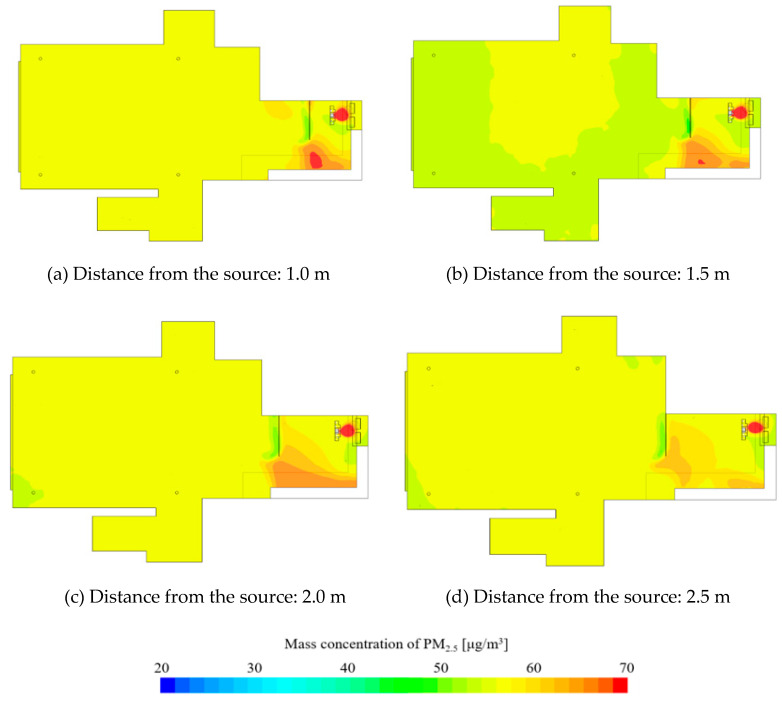
PM 2.5 concentration of horizontal section.

**Figure 9 ijerph-17-07799-f009:**
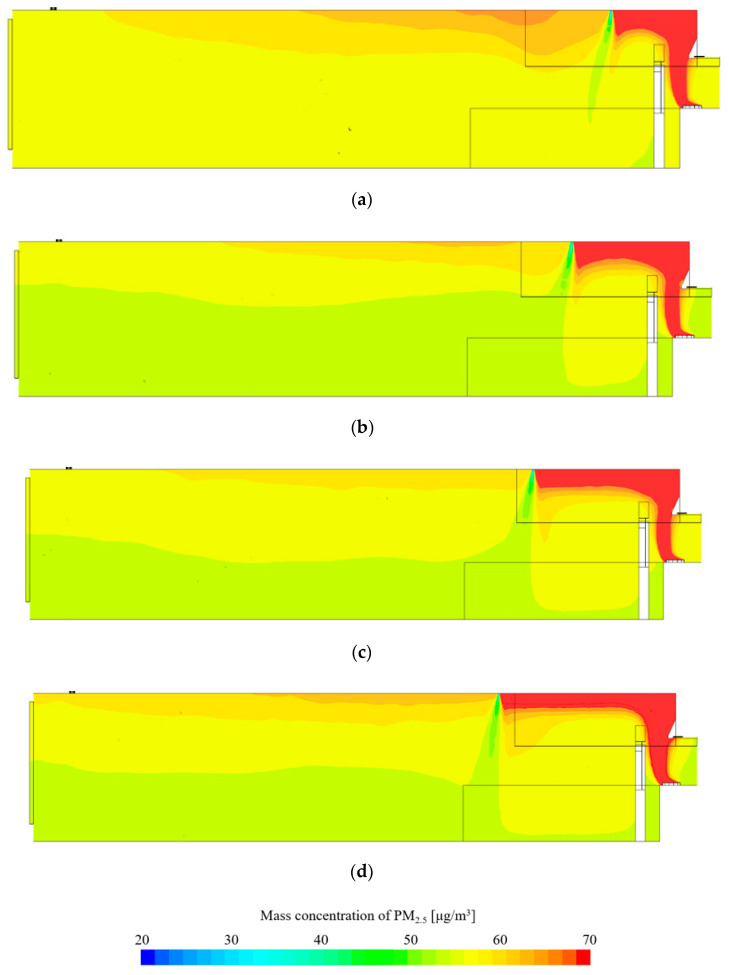
PM 2.5 concentration of vertical section. (**a**) Distance from the source: 1.0 m; (**b**) distance from the source: 1.5 m; (**c**) distance from the source: 2.0 m; (**d**) distance from the source: 2.5 m.

**Figure 10 ijerph-17-07799-f010:**
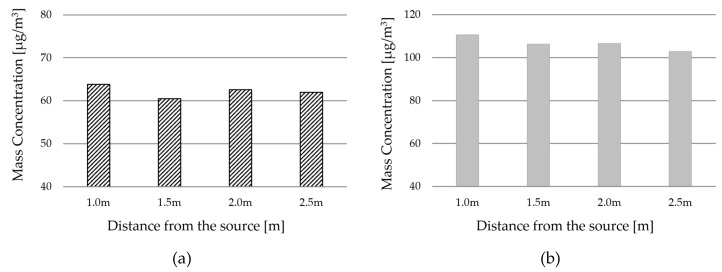
PM concentration of living room by distance from the source: (**a**) concentration in the living room; (**b**) concentration in the kitchen.

**Table 1 ijerph-17-07799-t001:** Boundary conditions for CFD simulation.

Boundary	Boundary Condition	Value
Auxiliary supply diffuser	Velocity inlet	Surface temperature: 27.5 °C Velocity: 1.929 m/s (150 CMH)
Range hood outlet	Pressure outlet	Pressure: −10 Pa
Frying pan (surface)	Wall (heat, emission source)	Surface temperature: 163 °C Emission rate of PM2.5: 3.068 mg/s
Wall	Wall (Temperature)	Surface temperature (floor, sink, walls of kitchen and living room): measurement data Other surfaces: adiabatic
Occupant	Wall (heat source)	Heat: 76 W

**Table 2 ijerph-17-07799-t002:** Information for the grid sensitivity analysis.

	Coarse Grid	Medium Grid	Fine Grid
Minimum size (mm)	2	1	1
Target size (mm)	200	100	80
Cells	393,515	583,371	1,109,294

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
