# Peer review of "CFD Simulation Analysis on Make-up Air Supply by Distance from Cookstove for Cooking-Generated Particle"

_ijerph, 2020, doi:10.3390/ijerph17217799_

Round 1
Reviewer 1 Report
This paper describes a case study on the CFD simulation analysis on make-up air supply by distance from cookstove for cooking-generated particles in Korean typical house. The simulation results provided useful suggestions on the way of installation of the auxiliary supply system. However, several points should be justified before acceptance.
- The CFD simulation was conducted when the ventilation system was deactivated. As authors described, the effect of auxiliary supply system can be singled out under this condition. However, are there any persons who actually dare to cook in the house without any ventilation? I believe this CFD simulation should be conducted when the ventilation system is on. Please provide more comments or description on this point.
- As described in Page 3, line 73-74, PM2.5 concentration was measured for validation of the CFD simulation. However, I can not find the result in this manuscript. Why omitted?
- In page 7, line 167, authors described the temperature above the cookstove was kept constant at 163 degC. My simple question is whether the temperature is enough to produce PM from foods. For example, the smoke temperature (smoking point) of beef tallow is 220 degC. Please explain why authors fixed the temperature at 163 degC.
- The CFD model developed in this study seems to work well for simulations of CO2 and temperature in the house, as described by authors. However, behavior of PM in indoor environment is quite different with that of CO2. The PM does not simply emit from food and spread in indoor air. As for emission, secondary formation of PM from gaseous components should be considered because the gases emitted form cooking are cooled in indoor air to form particles. While the CO2 is almost inert to interior materials, PM tends to sink on the ceiling or walls by adsorption. Have authors considered such differences between CO2 and PM in this simulation?
- Highlight of the research results or conclusion should be included in abstract.
Author Response
We are grateful to the reviewers for a very thorough review and constructive comments. The comments are listed below with our responses. We have revised the manuscript based on reviewers’ comments. We hope that our responses and the changes in the revised manuscript are sufficient for the reviewers.
- As described in Page 3, line 73-74, PM2.5 concentration was measured for validation of the CFD simulation. However, I can not find the result in this manuscript. Why omitted?
|
Response: |
|
PM2.5 concentrations were measured to establish emission rate for CFD simulations. The measuring equipment was installed in the duct outlet of the range hood. Measurement data were averaged to obtain the emission rate. To illustrate this, we added the description to Chapter 3.2. (line 153-158, 161-162) |
In page 7, line 167, authors described the temperature above the cookstove was kept constant at 163 degC. My simple question is whether the temperature is enough to produce PM from foods. For example, the smoke temperature (smoking point) of beef tallow is 220 degC. Please explain why authors fixed the temperature at 163 degC.
|
Response: |
|
As the reviewer said, it is cooked at a higher temperature during the actual cooking process. However, the temperature of the cookstove and the temperature of the top of the cooking pot are different. CFD simulations require the upper surface temperature of the pot and frying pan. n this study, the upper temperature of the frying pan was measured, so it was somewhat different from the actual smoke temperature. To explain this, line 87 was modified and Chapter 3.2 was added. (line 159-161) |
The CFD model developed in this study seems to work well for simulations of CO2 and temperature in the house, as described by authors. However, behavior of PM in indoor environment is quite different with that of CO2. The PM does not simply emit from food and spread in indoor air. As for emission, secondary formation of PM from gaseous components should be considered because the gases emitted form cooking are cooled in indoor air to form particles. While the CO2 is almost inert to interior materials, PM tends to sink on the ceiling or walls by adsorption. Have authors considered such differences between CO2 and PM in this simulation?
|
Response: |
|
The dynamic behavior of PM are strongly related to particle size. And Most of the PM generated during cooking has a particle size of 1μm or less. Therefore, cooking-generated particle is less affected by gravity. Nevertheless, multi-phase models should be used to analyze fine dust. To explain this, we added the description to the line 87-88 and section 3.1. (line 118-125) |
4 Highlight of the research results or conclusion should be included in abstract.
|
Response: |
|
We added a sentence to describe the results of the study in abstract. (line21-22) |
Reviewer 2 Report
This paper presented a computational fluid dynamics (CFD) study on the influence of the variant locations of the supply diffuser on the spread of cooking-generated particular matter (PM). The CFD simulations were performed based on the Reynolds-averaged Navier-Stokes (RANS) turbulence model, and validated against experimental measured room temperatures at different measurement locations and carbon dioxide (CO2) concentrations. Lastly, suggestions on the optimal installation location of the supply diffuser were provided. Overall, this paper is fairly well-written with a moderate amount of analysis and discussions. The conclusion is solid and well supported by the results. I recommend publishing this paper after the following issues being resolved.
- Please consider improving the resolution of several figures, as their current resolution may be slightly lower for publication purpose. They are Figures 1, 5, 6, 7, and 10.
- Please provide more description on Figure 2. For example, what does P1, P2, and P3 stand for?
- In Section 2.2, does the CO2 concentration measured at the same locations where temperatures were measured? Please briefly describe the CO2 measurement location(s) in this section.
- In Section 3, please briefly describe the boundary conditions specified for the CFD simulation.
- The sentence in Line 162-163 seems to be problematic. How to compare the temperature data WITH carbon dioxide concentration data? Do the authors mean “simulation was validated by comparing simulated room temperatures and carbon dioxide concentrations with the experimental measured data?”
- In Figure 6, does each experimental data at a given height represent an averaged value of temperatures taken at different wall locations at the same height? Could the authors provide error bars of the experimental data? Moreover, please explain in the text, why the discrepancies of simulated and measured data becomes larger at higher height?
- Similarly, in Figure 7, if CO2 concentrations were measured at multiple locations in kitchen and living room, please indicate the value (bar) presented is an average (or some other statistical quantities) and provide error bars.
- Please center the caption of Figure 8.
Author Response
We are grateful to the reviewers for their very thorough review and constructive comments. The comments are listed below with our responses. We have revised the manuscript based on the reviewers’ comments. We hope that our responses and the changes made in the revised manuscript are sufficient.
- Please consider improving the resolution of several figures, as their current resolution may be slightly lower for publication purpose. They are Figures 1, 5, 6, 7, and 10.
|
Response: |
|
We have revised all figures in the manuscript, including figure 1,2,6,7 and 10. |
Please provide more description on Figure 2. For example, what does P1, P2, and P3 stand for?
|
Response: |
|
First, there was an error in figure2. The location for P3 in figure2 has been modified. We added some descriptions to help understand the measurement points. (line 68-71) |
In Section 2.2, does the CO2 concentration measured at the same locations where temperatures were measured? Please briefly describe the CO2 measurement location(s) in this section.
|
Response: |
|
It may be partially explained as a supplement to previous modification. (line 68-71) In addition, we have added a description of the measurement height. (line 92-93) |
In Section 3, please briefly describe the boundary conditions specified for the CFD simulation.
|
Response: |
|
We agree with the opinion that there is not enough explanation for the boundary conditions. We added Chapter 3.2 to explain the boundary conditions for the CFD simulation. (line149-164) |
- The sentence in Line 162-163 seems to be problematic. How to compare the temperature data WITH carbon dioxide concentration data? Do the authors mean “simulation was validated by comparing simulated room temperatures and carbon dioxide concentrations with the experimental measured data?”
|
Response: |
|
The description of grid sensitivity was insufficient in the existing manuscript. To perform the grid sensitivity analysis, we compared the temperature and the carbon dioxide concentration data of the fine grid model. Each data was normalized by the data in the fine grid model. I supplemented the explanation on line 176-179. |
- In Figure 6, does each experimental data at a given height represent an averaged value of temperatures taken at different wall locations at the same height? Could the authors provide error bars of the experimental data? Moreover, please explain in the text, why the discrepancies of simulated and measured data becomes larger at higher height?
|
Response: |
|
The description of the existing manuscript was somewhat difficult to understand. The temperature data for figure 6 is the average temperature for each point. And the temperature was measured in the center of the living room and occupant position in the kitchen not in the wall locations. We have added a description for this on line 80-83. We added error bar to figure 6. And we added a description of the differences between simulation data and measurement data as they get higher. (line 191-201) |
- Similarly, in Figure 7, if CO2 concentrations were measured at multiple locations in kitchen and living room, please indicate the value (bar) presented is an average (or some other statistical quantities) and provide error bars.
|
Response: |
|
We added error bar to figure 7. And we added a description of figure7. (line 203-205) |
- Please center the caption of Figure 8.
|
Response: |
|
The caption of Figure 8 has been modified to the center |
Round 2
Reviewer 1 Report
The manuscript has been properly corrected considering reviewer's comments.